# Unified Probabilistic and Similarity-Based Position Estimation from Radio Observations

**DOI:** 10.3390/s25134092

**Published:** 2025-06-30

**Authors:** Max Werner, Markus Bullmann, Toni Fetzer, Frank Deinzer

**Affiliations:** 1Center for Artificial Intelligence (CAIRO), Technical University of Applied Sciences Wuerzburg-Schweinfurt (THWS), 97082 Wuerzburg, Germany; max.werner@thws.de (M.W.);; 2cronn GmbH, 53227 Bonn, Germany

**Keywords:** indoor, localization, fingerprinting, similarity, density estimation, particle filter

## Abstract

We propose a modeling approach for position estimation based on the observed radio propagation in an environment. The approach is purely similarity-based and therefore free of explicit physical assumptions. What distinguishes it from classical related methods are probabilistic position estimates. Instead of just providing a point estimate for a given signal sequence, our model returns the distribution of possible positions as continuous probability density function, which allows for appropriate integration into recursive state estimation systems. The estimation procedure starts by using a kernel to compare incoming data with reference recordings from known positions. Based on the obtained similarities, weights are assigned to the reference positions. An arbitrarily chosen density estimation method is then applied given this assignment. Thus, a continuous representation of the distribution of possible positions in the environment is provided. We apply the solution in a Particle Filter (PF) system for smartphone-based indoor localization. The approach is tested both with radio signal strength (RSS) measurements (Wi-Fi and Bluetooth Low Energy RSSI) and round-trip time (RTT) measurements, given by Wi-Fi Fine Timing Measurement. Compared to distance-based models, which are dedicated to the specific physical properties of each measurement type, our similarity-based model achieved overall higher accuracy at tracking pedestrians under realistic conditions. Since it does not explicitly consider the physics of radio propagation, the proposed model has also been shown to work flexibly with either RSS or RTT observations.

## 1. Introduction

Setting up indoor navigation requires extensive information about the individual circumstances of an environment and the behavior of the agent. To successfully guide agents such as people or robots through buildings, it may not be sufficient to rely solely on an absolute positioning system, such as GNSS [1], to track their trajectories over time. It is indeed possible to achieve errors of just a few meters this way. However, the goal should rather be to create a system that takes into account the logical structure of the building. Depending on the requirements and the context, a position estimate that is off by 4 m might still be considered sufficient, as long as it still suggests the correct room. But estimating either a different room, the wrong floor, or even a practically inaccessible area, should be considered undesirable behavior.

### 1.1. Background

Since navigation is a temporal process until a desired destination is reached, position estimation should also be considered in a temporal context. Based on the known current position of an agent, the range of possible positions at the next time step can be narrowed down based on the agent’s motion speed, direction, and other behavioral information. Further movement constraints are given by the building layout and obstacles. Under normal circumstances, it is rather unlikely that an agent will move straight through a solid wall or change floors without using stairs or an elevator. Such aspects can be taken into account by integrating a building plan into the position estimation.

It is still useful to also have an absolute positioning component that is not bound to the temporal context. This way, a start position can be given if there is no other prior information available. Relative motion predictions can also be validated and corrected if necessary by using absolute positioning [2]. While vision-based positioning with optical sensors or cameras is commonly used in robotics, our work focuses on data that can be captured by using standard smartphones without additional hardware. The idea is to provide a convenient solution for pedestrians to locate and navigate indoors. For this purpose, the information source of interest is the radio propagation inside a building. Both the measurable strength and the travel time of a signal are physically bound to the distance between transmitter and receiver. Knowledge about multiple transmitters can therefore help to determine the position from which the signals are received [3]. An obvious choice for transmitters to consider are pre-installed Wi-Fi access points (APs), but also custom installations such as Bluetooth Low Energy (BLE) beacons are an option. Radio observations are affected by measurement noise and irregularities in the signal propagation due to obstacles such as walls or furniture. These effects should be considered when estimating positions based on such observations.

### 1.2. Scope

We propose a probabilistic, similarity-based model for estimating the position of a pedestrian within a mapped indoor environment from radio observations. For that, we consider RSS values, given by the Wi-Fi or BLE RSSI in dBm, and round-trip time (RTT) values, given by Wi-Fi Fine Timing Measurement (FTM) distances in mm. We name it ObsSim, which stands for observation similarity. The model does not explicitly make any assumptions on physical aspects of the signal propagation. Instead, it uses the similarity of incoming AP signal patterns to reference observations to assign weights to the corresponding known positions. The probabilistic property of the model mainly refers to the outputs which are represented as probability density functions. We chose this realization to both accommodate to the uncertainties in the observations and to provide a continuous representation from which prior estimates can be evaluated. We conduct experiments to compare the capabilities of the proposed model with existing, physical distance-based approaches, and to identify differences between the two concepts in terms of estimation accuracy.

### 1.3. Related Work

In the following, we provide an overview of the State-of-the-Art in indoor localization techniques. First, we present related work on similarity-based approaches to localization via RSS measurements. After that, we focus on work on RTT and its purposes for localization, since we want to support this type of measurement with our proposed model. The last paragraph is dedicated to work on integrating such models into complete fusion approaches. Here, radio measurements are combined with other available sensor data to provide reliable localization over time, which is also the applied goal for our model.

Techniques for positioning based on the radio propagation in an environment have been studied for several decades. Most solutions refer to RADAR, the work of Bahl and Padmanabhan from 2000 [4]. They propose RSS-based position estimation by comparing incoming radio observations with reference data consisting of observations that are mapped to known positions. To describe this type of reference data, the term fingerprints (FPs) has been established in the research field. The approach focuses on the similarity of information provided by the FPs. For each FP, a signal vector that contains the average RSS per AP is created. With a k-nearest neighbors (k-NN) search, they obtain the most similar FPs via an Euclidean distance measure. The weighted average is then used to calculate a position estimate. A comprehensive study was made by Torres-Sospreda et al. in 2015 [5]. They analyzed alternatives to the well-established Euclidean distance as similarity measure for the k-NN. In addition, they tested adjustments to the representation of RSS data which is typically given in dBm and therefore on a logarithmic scale. In the experiments, they found that using a Sørensen distance measure along with a exponentiated representation of the RSS values could significantly improve localization accuracy over the Euclidean distance measure. Sun et al. developed an approach to reduce the effort of RSS fingerprinting in 2018 [6]. Using a Gaussian Process Regression, they augment FPs based on the given data. Different variations in the implementation could improve the localization accuracy compared to using the original FPs without augmentation. In an evaluation scenario with a k-NN-based algorithm, they improved a localization error from 6.9 m to 5.1 m in the best configuration with augmented FPs. The selection of appropriate hyperparameters is crucial for any predictive model. A useful extension to earlier k-NN approaches was given by Torres-Sospreda et al. in 2024 [7]. They propose a pipeline to estimate the right number of neighbors and distance or dissimilarity function. Although the accuracy improvement achieved is not significant, the authors highlight the potential of such adaptive approaches.

RSS is not the only metric that can provide information about the radio propagation. With the introduction of the Fine Timing Measurement (FTM) procedure into the Wi-Fi specification in 2016 [8], distance measurements over RTT were enabled for consumer devices. Since then, indoor positioning using FTM has been subject to several studies. Gou et al. [9] analyzed the FTM ranging in 2019 and proposed a calibration method to minimize the measurement offset that is caused by processing time delays during the RTT procedure. In experiments for smartphone-based localization, they achieved an improvement from 3.4 m with RSS fingerprinting to 2.0 m for their calibrated RTT ranging method. The properties of FTM and RSS in the localization context were compared side by side by Bullmann et al. in 2020 [3]. Despite FTM being conceptually less prone to obstacle attenuation than RSS, a similar error behavior could be assessed, due to additional propagation side effects that persist with FTM. To achieve better accuracy compared to RSS, they suggested an offset as well. An FTM approach by Horn [10] from 2024 uses an inverse Bayesian grid method to perform outdoor localization in urban outdoor spaces, close to the Wi-Fi infrastructure of buildings. In experiments with smartphones, the average localization error was reduced from 5.9 m to 4.2 m compared to standard GNSS.

Modern indoor pedestrian localization systems do not only rely on radio measurements. They also perform sensor fusion with sensors such as the accelerometer, gyroscope, or magnetometer from an inertial measurement unit (IMU) for Pedestrian Dead Reckoning (PDR) [11]. The additional information can help to increase the localization stability and accuracy when tracking pedestrians over time. A common approach is to use a probabilistic Bayes Filter framework for recursive state estimation. Several approaches are based on a Kalman Filter (KF), e.g., Chen et al. (2015) [12], Guo et al. (2019) [9], and Liu et al. (2021) [13]. Approaches using a Particle Filter (PF) were brought up by Harle (2013) [14], Fetzer et al. (2018) [15], and Shen et al. (2020) [16]. The position estimate at the current time step of such a filter is contained in the current state estimate. During each recursive step, a relative state transition based on PDR is performed. These transitions can adapt naturally to the pedestrian’s motion, but they would require a known starting position, and due to imperfections in the sensor data, state estimates are likely to drift apart over time. The role of the radio sensor model is to provide absolute positioning to evaluate and correct state estimates. For this purpose, the model must be able to evaluate state estimates at any position. This requirement makes the use of distance or propagation models intuitive, which is why they are often used and preferred over similarity-based models in the recent literature. The integration of distance models is rather straightforward, as they can be used to directly evaluate state assumptions by comparing measurements for each AP with the signal strengths that are predicted by the model. Related approaches for the fusion integration of similarity-based models are given by Zou et al. (2017) [17] and Nguyen et al. (2018) [18], each with rather simplified assumptions of position distributions that fit observed radio signals. In our proposed work, however, we aim to pay more attention to the underlying distribution of possible position estimates by introducing density estimation.

### 1.4. Outline

In Section 2, we explain our existing indoor localization system in which the model is integrated. The explanation of the model itself is given in Section 3. The experiments we conducted with the model in realistic scenarios, are documented in Section 4. In Section 5, we provide a summary and suggest future work.

## 2. Context

Before presenting the methodology, we refer to our existing work on the localization system into which the proposed ObsSim model is integrated. Even though it is conceptually feasible to use the ObsSim model as a standalone, we aim to apply and evaluate it in the context of a full sensor fusion approach.

### 2.1. Existing Particle Filter System

The ObsSim model is set to be integrated into our existing smartphone-based indoor localization solution, which is explained comprehensively in [15]. The recursion scheme is given by(1)$$p\left(q_{t}|o_{1:t}\right)\propto \underset{\begin{array}{c} \mathrm{evaluation}\\ \mathrm{sensor}\;\mathrm{model}\\ \left(\mathrm{scope}\;\mathrm{of}\;\mathrm{this}\;\mathrm{work}\right) \end{array}}{\underset{︸}{p(o_{t}|q_{t})}}\int \underset{\begin{array}{c} \mathrm{transition}\\ \mathrm{motion}\;\mathrm{model} \end{array}}{\underset{︸}{p(q_{t}|q_{t-1},o_{t-1})}}\underset{\mathrm{recursion}}{\underset{︸}{p(q_{t-1}|o_{1:t-1})}}\;dq_{t-1},$$
where $q_{t}$ denotes the current state at the time step *t* and $q_{t-1}$ the state at the previous time step. Accordingly, the current observation is $o_{t}$ and a previous one given by $o_{t-1}$. All observations in total are denoted by $o_{1:t}$ and all previous ones by $o_{1:t-1}$. For state transitions, a PDR motion model is used. The PF is realized with the CONDENSATION algorithm [19]. Here, a finite set of weighted particles, each representing an assumption of $q_{t}$, surpasses the transition first and is then evaluated using the sensor model, which assigns a weight to each particle. Before the next transition, the particles are resampled based on their weights to ensure that they are always concentrated near the highest current probability mass of the filter. The (weighted) average of all particles is always considered to summarize the current estimate.

In the reference system, $q_{t}$ consists of(2)$$q_{t}={\left(x,y,z,\theta \right)}_{t},$$
with *x*, *y*, and *z* representing the 3D coordinates of the pedestrian, and θ the absolute heading [15]. The information from the smartphone’s sensors at a time is represented by(3)$$o_{t}={\left(s_{\mathrm{radio}},\Delta \theta ,n_{\mathrm{steps}},\Omega \right)}_{t},$$
with $s_{\mathrm{radio}}$ being all currently observed radio measurements from reachable APs, Δθ the detected heading change from the IMU, *n*_steps_ the number of steps counted during the last PF interval, and Ω the result from an activity recognition [15]. The latter uses the IMU to determine whether the pedestrian is either standing, walking straight, walking upstairs, or walking downstairs.

An essential component of the system is the floor plan, which provides information about the layout of the building. Not only does it help users to orient themselves, but it also provides useful context to the PF. Based on walls and other types of obstacles, a triangular navigation mesh is spanned through the environment [15]. The mesh determines the “walkable” area on which particles are allowed to move. This way, transitions can only be made through adjacent triangles and particles are prevented from intersecting obstacles.

The ObsSim model we propose in this paper is represented by the evaluation term from the recursion scheme in (Equation 1). It is entirely based on the measurements from $s_{\mathrm{radio}}$, which can be either Wi-Fi, BLE, or both.

### 2.2. Optimized Log-Distance Model

Before going into details for the new model, we refer to our existing optimized Log-Distance model for RSS data. It is a propagation model that uses a modified variant of Frii’s path loss equation [20] for calculating distances based on RSS. The modified equation for the RSS μ at a known position ρ is given by [15].(4)$$\mu_{\rho }=P_{0}-10\gamma \log_{10}\frac{d}{d_{0}}+\Delta f\beta$$
Here, $P_{0}$ is a reference RSS measured at a distance $d_{0}$ (usually 1 m) and β is a constant attenuation factor. Δf marks the number of floors or ceilings, respectively, between the transmitter and the receiver. Walls and other obstacles could also be considered for the attenuation [21]. However, since they would impact the computational effort much more, they are omitted here. For each AP, the parameters $P_{0}$, γ and β are optimized together with the position of the AP. This way, it is ensured that the signal propagation of each AP is represented best by the corresponding function. When applying the function, *d* and Δf can be derived by knowing ρ, which is contained in q. If the AP position within the map is known, it is accepted as a prior but still included in the optimization process. To evaluate a particle and update its weight, a Gaussian distribution with a user-defined noise variance $\sigma^{2}$ is assumed with the RSS function value for each AP. The current incoming measurement from each AP is then evaluated against this function.

### 2.3. Binned Skew Normal FTM Model

In the case of Wi-Fi FTM data, we used a different approach in the past. The measurements are already distance estimates, but with their own special characteristics. With the Binned Skew Normal FTM model [22], it is assumed that the FTM distances for the same actual distance to an AP are distributed according to a skew normal distribution SN(ξ,ω,α). Compared to a standard normal distribution, ξ is the location parameter corresponding to the μ, and ω is the scale parameter corresponding to $\sigma^{2}$. The additional shape parameter α controls a tail of the density function that is either to the left from ξ for α<0 or to the right for α>0.

When optimizing the model, functions are fitted that describe the three parameters of the distribution, given the well-known distance to the AP. To achieve this, the measurements from the FPs are first divided into bins for each m of actual distance to then fit functions for each parameter given the current distance. A first-order polynomial $\hat{\xi }\left(d\right)$ is fitted for ξ, given a distance *d*. For $\hat{\omega }\left(d\right)$, a second-order polynomial is fitted and another first-order polynomial for $\hat{\alpha }\left(d\right)$. To now assign a weight to a particle of the PF according to its well-known distance *d* to an AP and the measured FTM distance $d^{\mathrm{ftm}}$, the probability density function is given by(5)$$p\left(d^{\mathrm{ftm}}|d\right)=\frac{2}{\hat{\omega }\left(d\right)}\varphi \left(\frac{d^{\mathrm{ftm}}-\hat{\xi }\left(d\right)}{\hat{\omega }\left(d\right)}\right)\Phi \left(\hat{\alpha }\left(d\right)\frac{d^{\mathrm{ftm}}-\hat{\xi }\left(d\right)}{\hat{\omega }\left(d\right)}\right),$$
where φ(·) is the density function of a standard normal distribution with zero mean and Φ(·) is the cumulative distribution function of the standard normal distribution.

## 3. Proposed Methodology

This section provides a detailed description of our proposed ObsSim model for radio-based position estimation. Since the approach is purely similarity-based, the data must first be processed to make the radio measurements comparable. This also involves proper sparsity handling for the case that measurements are incomplete or missing. Central to the approach is the inference of density functions for position estimates, based on the similarities from the incoming observations with the signal pattern of each FP. In the following, the data, preprocessing and model design are explained.

### 3.1. Fingerprints and Radio Observation Preprocessing

Reference data for radio-based indoor localization is commonly known by the term fingerprints (FPs). From a machine learning perspective, where the sensor model can be formulated as a regression problem, the term training data would also be applicable. In the context of this model, however, FPs are more used as a lookup table, on which no training is actually done. In an FP dataset, known positions are mapped to radio observations from reachable stationary transmitters, usually access points (APs).

In order for the position estimation to work properly, the entire environment should ideally be covered by APs in such a way that multiple APs can be reached simultaneously from any position. FP positions should be spread over the entire walkable area of the environment to gain a solid overall impression of the radio propagation. In Figure 1, a possible setup is shown.

A suitable data representation is required to compare incoming observations with those represented by the FPs. The position of a FP from the dataset and therefore the model target is given by ρ=(x,y,r). Here, *x* and *y* represent the 2D coordinates within the environment’s coordinate grid and *r* is a region index. We denote the set of all regions by *R*. A region can contain either a floor or a certain area of the building at a consistent ground height. For the vertical component of the localization, position estimates should be focused on the structural layout of the building and less on the exact Cartesian coordinates. Therefore, determining the correct region is usually more important than the absolute height, which can still be derived from the region if desired. Thus, the model is kept simple by sticking to a discrete index representation. The complete set of FP positions is given by(6)$$P=\left(\begin{array}{cccc} x_{1} & y_{1} & \ldots & r_{1}\\ x_{2} & y_{2} & \ldots & r_{2}\\ \vdots & \vdots & \ddots & \vdots \\ x_{n_{\mathrm{fp}}} & y_{n_{\mathrm{fp}}} & \ldots & r_{n_{\mathrm{fp}}} \end{array}\right)\;.$$

FPs are recorded by scanning for APs with a smartphone at each FP position for approximately 20 s to 30 s. During this period, multiple measurements can be expected from each AP, as shown in Figure 2. The distribution of the values is affected by measurement noise and other physical aspects like multipath propagation. Recent related approaches rely more on modeling the distribution of measurements per AP and treating all APs individually. But assuming the radio propagation of each AP to be statistically independent can be inappropriate. Considering correlations between multiple APs may deliver more useful information about the characteristics of an environment. For our proposed approach, we stick to a rather simplified view of the individual measurement distributions. Instead, we rely more on combined signal patterns over multiple APs, without explicitly considering the physical properties of radio propagation. The FP observations of our model are stored in a matrix given by(7)$$M=\left(\begin{array}{cccc} m_{1,1} & m_{1,2} & \ldots & m_{1,n_{\mathrm{ap}}}\\ m_{2,1} & m_{2,2} & \ldots & m_{2,n_{\mathrm{ap}}}\\ \vdots & \vdots & \ddots & \vdots \\ m_{n_{\mathrm{fp}},1} & m_{n_{\mathrm{fp}},2} & \ldots & m_{n_{\mathrm{fp}},n_{\mathrm{ap}}} \end{array}\right),$$
where $n_{\mathrm{fp}}$ is the number of FPs, $n_{\mathrm{ap}}$ is the number of known APs, and $m_{i,j}$ contains one or many feature aggregations of the radio measurements at the *i*th FP and *j*th AP. The aggregations can be combinations of the mean or median or other numerical features of the measurement value distribution. To maintain consistency, each $m_{i,j}$ must contain the same number and order of aggregations.

FPs are collected by simply standing at certain distinct positions. For the localization use case, however, the proposed model must be able to also estimate positions from walking pedestrians. Individual radio scan results can arrive at virtually any time. Therefore, in order to pass a suitable amount of data to the model at the same time, it is necessary to summarize measurements over a certain timespan. Our method is based on the 500 ms update interval of the PF. We introduce a moving time window, which has a size Δt equal to or greater than the update interval. Choosing an optimal Δt poses a dilemma: Smaller time windows contain only very recent measurements, but this data may be too little to produce meaningful position estimates. Larger time windows can provide more information, but some may be taken from a position that is too far away from the pedestrian’s current position. We chose to optimize the model to work with a window size that is equal to the 500 ms update interval.

Once the model is deployed for actual localization, an observation vector $m^{\prime }$ is created out of all measurements from a time window, while maintaining the AP and feature aggregations from the previously constructed M. Therefore, $m^{\prime }$ has the same structure as one row from M. In case of missing values, the corresponding fields are left empty.

### 3.2. Observation Comparison with a Kernel Function

The proposed ObsSim model is built upon the assumption that(8)$$m^{\prime }∼M_{i,:}\Rightarrow \rho^{\prime }∼P_{i,:}\;.$$
As similarity measure between a given $m^{\prime }$ and $M_{i,:}$, we use a kernel that is inspired by covariance functions for Gaussian Processes. A common example is the squared-exponential or radial basis function kernel (RBF), which is given by [23].(9)$$k_{\mathrm{RBF}}\left(\tau \right)=\exp\left(-\frac{\tau^{2}}{2l^{2}}\right)$$
The length-scale parameter *l* controls how different two compared values can be considered similar. In this model, $\tau =∥m^{\prime }-M_{i,:}∥$. By applying the kernel on each row of M to compare it with $m^{\prime }$, a similarity vector k of length $n_{\mathrm{fp}}$ is constructed. For the *i*th FP in the model and therefore the *i*th element in k, $k_{i}$ determines the probability that $m^{\prime }$ was observed somewhere near the position $P_{i,:}$. An example for this step is given in Figure 3, where the highest similarities are assigned nearby the position where the actual observation was made. The use of a kernel as similarity measure is different from most established models that use a Euclidean distance measure, which represents a dissimilarity instead [4,5,24]. However, for the upcoming steps of our approach, it is more practical to have an actual similarity measure. When it comes to weighting particles of a PF, a high similarity should correspond to a high weight, indicating a high probability of the particle’s state.

At this stage, it is important to address sparsity. It cannot be guaranteed that every single element of $m^{\prime }$ or $M_{i,:}$ contains a value, since not every AP is available at all times and from all locations. Often, only a small fraction of the elements are present, making the data unsuitable for imputation strategies. Therefore, we decided to only use features present in both given vectors for the comparison. This way, the model will also not produce a position estimate if $m^{\prime }$ is empty. In this case, the PF would simply skip the evaluation procedure and rely solely on the transition model.

The example in Figure 4 shows that it can be counterproductive to always apply a kernel on intersecting features. Kernel similarities tend to be higher with fewer features, which would violate the concept of the model. Consequently, FPs with few considerable measurements could have a higher similarity than FPs that allow for a comprehensive comparison. To avoid this, we introduce an additional filtering strategy. It ensures that $k_{i}$ is only computed if most of the elements from $m^{\prime }$ are present in $M_{i,:}$. Otherwise, we set $k_{i}=0$. We require that the intersection contains at least 90 % of the APs in $m^{\prime }$. This is the case in Figure 4a, where only one out of 15 APs that is present in $m^{\prime }$ is unavailable in the current FP. In Figure 4b, however, the intersection is minimal, making the result useless. Thus, the similarity here is overwritten by 0.

### 3.3. Position Density Estimation

Based on the previously computed similarities, k, the sensor model must fit a probability density function that can be used in the evaluation step of the PF in (Equation 1) to assign weights to particles. To allow the use of classical inference methods for density estimation, we decided to generate a set of non-weighted samples $\hat{P}$ out of k and P. For this purpose, k is first normalized into a weight vector(10)$$w=\frac{k}{\sum_{i=1}^{n_{\mathrm{fp}}}k_{i}}$$
so that $\sum_{i=1}^{n_{\mathrm{fp}}}w_{i}=1$. With an arbitrarily selected sample count η, the number of samples drawn from each $P_{i,:}$ is given by(11)$$\eta_{i}=\left\lfloor w_{i}\cdot \eta +\frac{1}{2}\right\rfloor .$$
Since w contains decimal numbers, and each $\eta_{i}$ must be rounded to the nearest integer, some rounding errors are likely to occur, which causes(12)$$\hat{\eta }=\sum_{i=1}^{n_{\mathrm{fp}}}\eta_{i}\approx \eta .$$
However, if η is chosen large enough, e.g., in between 100 and 1000, the small remaining errors should be negligible. Otherwise, FPs that are so dissimilar that they cannot even be represented by a single sample, are filtered out entirely.

With the introduced sampling strategy, each position of P is simply sampled as often as the corresponding $\eta_{i}$ specifies. To smooth the distribution of $\hat{P}$ before inferring a density function, we introduce an optional sampling noise $\epsilon_{x},\epsilon_{y}∼N\left(0,\sigma^{2}\right)$. The *i*th sample in the set $\hat{P}$ based on $P_{i,:}$ is then given by(13)$$\hat{\rho }=\left(x_{i}+\epsilon_{x},y_{i}+\epsilon_{y},r\right).$$
When applying the sample generation on the example from Figure 3, $\hat{P}$ can be achieved as shown in Figure 5.

To apply density estimation on this sample set, it is important to treat each region, *r*, which a sample is assigned to, separately. In practice, an individual estimator must be fitted for every *r* in *R*, proportional to the corresponding number of samples ${\hat{\eta }}_{r}$. We realize the evaluation term from (Equation 1) by(14)$$p\left(o_{t}|q_{t}\right)=p\left(\hat{\rho }\right)=\sum_{r\in R}\delta_{r,r_{\hat{\rho }}}\frac{{\hat{\eta }}_{r}}{\hat{\eta }}p_{r}\left(\hat{\rho }\right)$$
to obtain the likelihood of a position $\hat{\rho }$. Virtually any density estimator that is suitable for inference can be applied. However, due to multipath effects and other influences on the observed radio propagation, estimates may either spread out into irregular shapes, or even concentrate around several distinct clusters. Therefore, our main interest lays on nonparametric estimators or mixture models.

#### 3.3.1. Kernel Density Estimation

A commonly used nonparametric model for density estimation is Kernel Density Estimation (KDE) [25]. Here, a kernel is assigned to each of the given samples so that their overlapping sum delivers a continuous representation of the sample density. Other than the RBF kernel from (Equation 9) that we use for observation similarity computation, a kernel *k* for KDE must itself be a valid density function. Thus, it needs to fulfill ∫k(τ)dτ=1. We use a Gaussian kernel, which is given by(15)$$k_{\mathrm{Gaussian}}\left(\tau \right)=\frac{1}{\sqrt{2\pi }}\exp\left(-\frac{\tau^{2}}{2}\right)$$
and essentially places a zero-centered normal distribution around each sample. When setting up the KDE for a specific *r*, the likelihood for $\hat{\rho }$ is given by(16)$$p_{r}\left(\hat{\rho }\right)=\frac{1}{{\hat{\eta }}_{r}h}\sum_{j=1}^{{\hat{\eta }}_{r}}k_{\mathrm{Gaussian}}\left(\frac{{\hat{\rho }}_{:2}-{\hat{P}}_{r;j,:2}}{h}\right),$$
whereby only the *x* and *y* coordinates are evaluated by the kernel. The bandwidth parameter *h* controls the scaling of each kernel, which influences how smooth the KDE turns out. In Figure 6, the application of KDE is visualized in our demonstrative scenario.

#### 3.3.2. Dirichlet Process Gaussian Mixture Model

The KDE can adapt well to the distribution of $\hat{P}$. In its nonparametric nature, however, it needs to consider every given sample for the evaluation when used within a particle filter. As an alternative option, we consider as well using a mixture model that fits $\hat{P}$ with only a few parameterized components. It is essentially a weighted sum of density functions that can be fully defined by just a few parameters [26]. The mixture density function for a model consisting of *J* components is given by(17)$$p_{r}\left(\hat{\rho };\theta \right)=\sum_{j=1}^{J}\pi_{j}p_{r;j}\left(\hat{\rho };\theta_{j}\right),$$
and it is fully defined by the parameter set $\theta :=\{\theta_{1},\pi_{1},...,\theta_{j},\pi_{j}\}$ [26]. It contains a non-negative weight that is given by $\pi_{j}$, for which applies $\sum_{j=1}^{J}\pi_{j}=1$. Furthermore, $\theta_{j}$ defines parameters of an individual component. In case of a Gaussian Mixture Model (GMM), where each component is a Gaussian normal distribution, $\theta_{j}=\left(\mu_{j},\Sigma_{j}\right)$ defines the mean vector and the covariance matrix. To fit a GMM on $\hat{P}$, θ must be optimized. A common method that only works for a user-defined *J*, is the famous expectation–maximization algorithm [27].

There is also a Bayesian approach that does not require the specification of *J*. The Dirichlet Process Gaussian Mixture Model (DPGMM) is conceptually an infinite mixture model [28]. Here, the weights of newly assigned components quickly converge to 0, so that a truncation leads to a managable number of components. To fit such a model in reasonable time during each recursive step of the PF, we consider an implementation that is based on variational inference [29]. Figure 7 shows the application of the DPGMM on our example for $\hat{P}$. Here, the partitioning of the distribution into weighted components is clearly visible.

## 4. Experiments and Results

To evaluate the capabilities of the ObsSim model, we perform experiments under realistic conditions. The model is deployed in three different configurations for position inference from the generated sample set: A simple, bivariate normal distribution, the KDE (see Section 3.3.1), and the DPGMM (see Section 3.3.2). Our evaluation is divided into two parts. The first part focuses on radio signal strength for Wi-Fi and BLE in two scenarios. The second part focuses on Wi-Fi FTM measurements in only a single scenario, where these measurements are available. Since we want to determine the capabilities of the new model compared to our established technology, we also include our existing optimized Log-Distance model into the experiments (see Section 2.2). For FTM, we include the Binned Skew Normal FTM model (see Section 2.3) and also add a simplified distance model. While the Log-Distance model for RSS is used to optimize functions to get from RSS to distances, the simplified model uses the provided FTM distances directly. There is no function optimization and also the AP coordinates from the floor plan remain unchanged.

### 4.1. Setup and Evaluation Criteria

To make the experiments application-oriented, we chose to always run it inside a PF with active PDR transition. The number of particles is set to 5000, and the interval over which the PF is updated and measurements are accumulated is set to 500 ms. To ensure that possible deficits of the radio model are not compensated directly within the evaluation step, the PF is either run with Wi-Fi or BLE measurements exclusively. We decided to also not include any further models beyond the radio model that would impact the evaluation step.

To prevent sample impoverishment in the PF, where very few particles receive a very high weight for resampling, we use a simple fallback mechanism. Without this, the localization could get stuck in situations with an inappropriate transition or evaluation. This workaround is especially necessary for elevators, which, unlike stairs, do not allow particles to change floors naturally in our current implementation. A second set of uniformly distributed recovery particles is evaluated the same way as the main particle set, but without applying transition or resampling. In each step, the three main particles with the lowest weights get replaced by the three recovery particles with the highest weights. This way, particles can be completely resampled to a new location if the PF loses track when no evaluation is applied anymore near the recent particle positions.

For the experiments, we consider recordings from two datasets, each from a different building with unique characteristics. In each of them, a pedestrian walks along a predefined ground truth (GT) path while holding a smartphone that continuously collects radio and IMU data. The GT is given as a series of waypoints within the known floor plan of each building. The pedestrian walks at a constant, natural speed and logs a timestamp each time a waypoint is passed.

The selection of hyperparameters can have a significant impact on the accuracy of the model. Therefore, we decided to manually adjust them for each model configuration and recording, so that each configuration is compared under the best individual conditions. To calculate a positioning error for each recursion step of the PF, the current GT position is interpolated based on the given timestamp. The current estimated position is given by the weighted average of each particle’s position. The absolute positioning error is based on the Euclidean distance between each estimated and GT position, so that(18)$$\mathrm{Error}=\sqrt{{\left(x_{\mathrm{estimated}}-x_{\mathrm{GT}}\right)}^{2}+{\left(y_{\mathrm{estimated}}-y_{\mathrm{GT}}\right)}^{2}+3\cdot {\left(z_{\mathrm{estimated}}-z_{\mathrm{GT}}\right)}^{2}}\;\left[m\right]\;.$$
Estimating an incorrect floor would be a practical problem that is not adequately reflected in the error metric, since the impact on the error is small in numbers compared to the horizontal error. Therefore, we multiply the vertical component by a penalty factor of 3 [15]. The vertical coordinate *z* is derived from *r* via the navigation mesh of the floor plan.

The PF does not behave deterministically due to its probabilistic transition and evaluation. Therefore, each run is repeated 10 times with the same model configuration for each recording. It is ensured that each repetition uses different seeds for all parts that involve random sampling. All metrics provided as experimental results refer to the average of the 10 runs. This reduces the effect of outliers on the results.

### 4.2. Experiment Scenarios

Our first environment consists of only one floor. The left half is an office space and the right half is a workshop. A total of 155 FP positions are arranged in a grid of about 1.5 m (see Figure 3). There are several obstacles such as office desks, workbenches, metalworking machines and storage racks. Dedicated for the purpose of the localization system, 23 ESP32-S3 microcontrollers are installed to act as APs for Wi-Fi and BLE.

For each FP position, there are measurements from three different smartphones: a Samsung Galaxy S21, a Google Pixel 2 XL, and a Google Pixel 3a.

For testing, a recording made with the Google Pixel 2 XL is considered. The GT path for each recording is shown in Figure 8. It starts in the workshop area, where the path first goes in a circle through the entire room, then through a narrow kitchen into the office area and finally returns back to the center of the workshop.

We use the university building from Figure 1 as our second experimental environment, with the GT path shown in Figure 9. For both the FPs and the recording, a Google Pixel 2 XL smartphone was used. This scenario is more challenging but also more realistic than the first one. There are 49 FP positions, which are much less densely distributed than in the first scenario. There are two floors that need to be correctly identified via their corresponding *r*. Especially challenging are floor changes, either by stairs or by elevator. The only available radio source is Wi-Fi from the building’s existing mesh. In total, 181 APs were identified by their MAC addresses in the FP dataset. The path starts on the top floor, at the longer end of the corridor, and continues almost the entire way until it turns into a dead end on the right. After a 180° turn, it passes a small office and a lecture room on the opposite side. It then re-enters the corridor and turns left into the more open, shorter segment of the corridor and crosses through a seating area. At the end of this area, a left turn leads down a staircase to the lower floor. A further 180° turn is made at the left end of the corridor, then the path passes another lecture room to get to the longer segment of the floor. After going around a floor cut-out, another lecture room and laboratory is passed. Then, the path goes to the longer end of the building into the elevator, which is taken upwards. Here, the path ends nearby the starting position.

### 4.3. Preliminary Analysis

Before discussing the results, we provide more information about the data from the test environments. This provides useful context for better interpretation of the experimental results.

The ObsSim model produces estimates based on the similarity of FP observations. A visualization of the pairwise similarities between the FP observations themselves for the data from scenario #1 is given in Figure 10. The diagonal grid pattern that is present for each signal is due to the grid scheme in which the FPs are ordered and enumerated row by row (see also Figure 3). The two well-defined clusters for Wi-Fi and BLE RSSI correspond exactly to the two large rooms into which the building is divided. This confirms the effect of walls attenuating the signal. There is no such clear separation in the FTM observations. Here, the effect of obstacles is still noticeable, but much less intense.

The distribution of FP measurements against the actual distances to the respective APs is plotted in Figure 11 for each measurement type. For Wi-Fi and BLE RSSI, the significant dip of the maximum values at about 20 m can be explained by the dimensions of the building. Measurements with distances beyond 20 m are therefore attenuated by at least the wall in the center of the building (see also Figure 8). Thus, above 20 m, the overall spread is also increased significantly for FTM. The smaller and visibly delimited sets of points that occur especially between 0 m and 20 m for the Wi-Fi data, are due to device-specific characteristics, since measurements from three different smartphones are included in the dataset. Remarkable is the consistent offset of about 50 m for FTM. The consistency of this error pattern is also present in the literature [9,22]. Processing delays in the AP add to the RTT and extend the calculated distance by almost always the same value. We are going to consider this effect during the experiment with the plain FTM distance model.

### 4.4. Discussion of the Results

In this section, we present the results of our experiments. The first part is focused on RSS observations, the second on RTT. After justifying the selection of hyperparameters, we provide diagrams and summary charts with comprehensive error metrics, and commentary on noteworthy effects.

#### 4.4.1. Radio Signal Strength for Wi-Fi and BLE

For ObsSim, convincing accuracy was reached by using an RBF kernel for observation comparison and a Gaussian kernel for the KDE. We found that the choice of kernels is less decisive than the selection of kernel parameters. For the feature aggregation, including both the mean and the median of measurements per AP turned out to be most promising. Beyond η=500, no significant improvements are noticeable; therefore, this number is chosen. The best remaining hyperparameters for each model configuration are shown in Table 1. Whereas, for the first scenario, ObsSim *l* and Log-Distance $\sigma^{2}$ are consistent over Wi-Fi and BLE, the smoothness differs for ObsSim between the signal types. When using the KDE, the smoothness added via $\sigma^{2}$ is practically overdrawn by *h*. Therefore, $\sigma^{2}$ is kept consistently low to adapt the smoothness via *h* instead. When using the DPGMM or the normal distribution, $\sigma^{2}$ is the only parameter that controls the smoothness. It is chosen larger for BLE than for Wi-Fi, just like *h* for the KDE. The second scenario has a lower density of FP positions and APs, which is seemingly compensated by choosing a greater *l*.

For the first scenario with Wi-Fi RSSI measurements, the errors are plotted over time in Figure 12. The large error at the beginning of the recording is due to a dropout of measurements. Since the particles are uniformly spread over the entire space, the PF has no indication of the absolute position during this time. Also, the first incoming observation leads to a significant peak after a few seconds, because the first position estimate is quite far off. But as soon as valid observations are available, the estimates converge close to the GT and are able to keep up well over the remaining time. The Log-Distance error is larger for a longer period (P1-P2). A possible explanation is that some crucial information from the FPs is lost during the optimization of distance functions per AP. While this model is minimalistic and robust, some details are lost that are still considered by ObsSim. The smallest error is between P2 and P3 where the path goes through a narrow kitchen and a small corridor to the more open office space.

A summary of the results can be found in Table 2. Compared to KDE and the normal distribution, which are almost equal, the DPGMM configuration for ObsSim is slightly less accurate. Here, multimodalities are low enough that it is sufficient to assume $\hat{P}$ to be normally distributed. While the normal distribution and the KDE can, therefore, fit the density very well, the DPGMM may split $\hat{P}$ into more components than necessary, which can cause some negative side effects.

The same recording also contains BLE observations, for which we set up models separately. The error over time is shown in Figure 13. Unlike for Wi-Fi, observations are already occurring at the beginning, so there is not such a significant peak here—at least for ObsSim. The rate at which BLE measurements are provided is also significantly higher than for Wi-Fi. As a consequence, the sensor model has more influence in the PF. The ObsSim error curves are again very similar, whereas the Log-Distance accuracy is visibly different and generally slightly worse.

Table 3 shows that the mean errors have improved for all models compared to Wi-Fi. For ObsSim, the choice of the density estimation has no effect except for small variations due to the randomness of the system. Log-Distance with BLE is about as good as ObsSim with Wi-Fi, but still about 0.6 m worse than ObsSim with BLE.

For the second scenario in the university building, where only Wi-Fi is available, the error curves are visualized in Figure 14. The optimization of the Log-Distance model results in a larger variance, which causes a significant error at the beginning, since the particles take a few PF updates until they are resampled correctly around the GT. Even then, over a wide stretch, Log-Distance is notably off whereas the ObsSim errors remain almost consistent. The stairs towards the lower floor are the first major obstacle in the recording, but they do not affect the accuracy of any configuration (P1). Between P1 and P2 in the rear area of the lower floor, Log-Distance is almost on par with ObsSim.

While the navigation mesh continues over stairs to allow for natural particle movement between floors, it does not extend to elevators in our implementation. The sensor model must be able to accurately resample particles to the current elevator position (P4/P5), which, in this particular scenario, applies to ObsSim with KDE.

A summary of the errors is given in Table 4. It is noteworthy that the overall accuracy is not much worse compared to scenario #1. This is despite the fact that the scenario is larger, more complex and has less data available. For the ObsSim model, assuming a normal distribution is the worst choice. The higher accuracy with KDE or the DPGMM confirms that the more complex building layout and lower FP density result in a more complex distribution of $\hat{P}$. In general, KDE turns out to be the best choice. It even maintains a relatively high localization accuracy during the elevator ride.

#### 4.4.2. Fine Timing Measurement for Wi-Fi

The basic configuration of the PF and the ObsSim model remains the same as in Section 4.4.1. The FTM experiment is only executed in scenario #1. The ObsSim hyperparameters from Table 1 are kept except for *l*, which is now set to 3000.0 to match the different scale of the FTM values which are processed in mm. The plain FTM distance model works best with $\sigma^{2}=5.0$ and a consistent offset of -50 m to shift the measurements close to the GT distance (see Section 4.3). For the Binned Skew Normal FTM model, the optimization results in the parameter equations(19)$$\begin{array}{cc} \hat{\xi }\left(d\right) & =1.31\cdot d+37.0\;\left[m\right], \end{array}$$(20)$$\begin{array}{cc} \hat{\omega }\left(d\right) & =-0.00756\cdot d^{2}+0.463\cdot d+3.58\;\left[m\right],\;\mathrm{and} \end{array}$$(21)$$\begin{array}{cc} \hat{\alpha }\left(d\right) & =0.00952\cdot d+0.847\;\left[m\right]. \end{array}$$
The constant of 37.0 m in $\hat{\xi }\left(d\right)$ combined with the positive 0.847 in $\hat{\alpha }\left(d\right)$ that transforms the shape further toward higher values, is consistent with the significant offset of FTM values that is shown in Figure 11.

A visualization of the accuracy over time for each model is shown in Figure 15. As with the Wi-Fi RSSI data (see Figure 12), there is a peak at the beginning for each model due to the measurements that start arriving after a few seconds of the recording. The clear separation of the three models in the first half is in line with the previously expected behavior. The plain FTM Distance model provides the worst generalization due to the superficial assumption of a static offset. The Binned Skew Normal FTM model takes into account the distribution of values depending on the true distance, and is therefore more accurate. By far the highest accuracy is achieved with the ObsSim model, with each density estimation method performing almost equally well. In the second half, which takes place on the other side of the building, all models behave differently. Here, all error curves are closer together, with the ObsSim model getting slightly worse and the FTM Distance model significantly better than before. This shows that the FTM Distance model is indeed able to achieve good results when the corrected measurements match the true distance, but this is obviously not always the case. As with Wi-Fi RSSI, the lowest overall errors are achieved between P2 and P3 and then after P4, in the narrow section of the building. The ObsSim model can generalize much better by focusing on the overall pattern of all reachable APs and not trying to estimate all distances independently.

The errors are summarized in Table 5. Compared to the previous results for Wi-Fi RSSI (see Table 2), all ObsSim results have improved significantly, but BLE is still even better (see Table 3). The Binned Skew Normal FTM yields slightly better results than the Log-Distance Model for Wi-Fi RSSI. The FTM Distance model without any further optimization is the worst model not only during the FTM experiment, but also compared to the RSS-based experiments.

In this experiment, the accuracy of ObsSim with KDE is slightly behind compared to the DPGMM and the normal distribution, showing that in this scenario, where the measurements are less affected by attenuation, a simple normal function provides the best representation of $\hat{P}$.

## 5. Conclusions and Future Work

In this article, we proposed a purely similarity-based approach for a radio sensor model that fits into a PF setup for smartphone-based indoor localization. In direct comparison with the best configurations for our existing distance-based models, the proposed ObsSim model was more accurate and consistent throughout all experimental scenarios.

While distance models rely heavily on an appropriate representation of independently evaluated observations from each AP, our ObsSim model considers a more generalized view on patterns that can be observed from all APs combined. This also makes the model flexible with respect to the used measurement types. It has been shown to work well on both RSS and RTT measurements. Here, the only critical parameter when it comes to supporting either the one or the other type of measurement, is the kernel length-scale parameter *l*. It must be set to match the numerical scale of the measurements to make the similarity information useful.

Regarding the density estimation performed after the similarity calculation, the ideal approach has shown to depend on environment-specific conditions. For an environment with densely distributed FP positions, where reference measurements have been made before, it is absolutely sufficient to assume a standard normal distribution that is fitted by maximum likelihood. KDE could be confirmed to be more useful when there is a lower overall density of FP positions, which is usually the case in larger buildings such as our second scenario. Here, the KDE was able to adapt to the more irregularly shaped position samples. The DPGMM, which is also capable of fitting multimodal samples, did not show any particular advantages over KDE in our experiments.

All in all, ObsSim is a robust and accurate radio sensor model. It achieves overall errors of less than 2 m under various conditions when deployed in a PF. When using BLE measurements, which are performed at a higher rate than Wi-Fi, the errors can even be close to 1 m.

For future improvements, we suggest some possible refinements based on our work.

There is no standard procedure for hyperparameter optimization for the ObsSim model. For easier deployment, it would be helpful to introduce methods to derive appropriate hyperparameters from the characteristics of the FP data. Furthermore, the selection of an appropriate density estimator may be automated, since the use of a standard normal distribution can be both better and computationally more efficient than the KDE in certain environments.

A full fusion approach would consider all the available data simultanuously. In our experiments, we considered each radio measurement type separately and focused on the individual signal properties. But it would be interesting to see if the ObsSim model could be improved by combining RSS and RTT measurements.

For the implementation of smartphone-based indoor localization in practice, the performance aspect is critical. Further investigations may focus on the computational efficiency and real-time capabilities on constrained mobile hardware.

In addition to a possible adaptive use of the model in an interacting multi-model PF, there is also room to improve the model itself by allowing it to adaptively adjust some hyperparameters during runtime. The variance of the resulting density function could be situationally adjusted based on confidence metrics about incoming observations. Such dynamic adjustments could also enable accurate localization in more dynamic environments where the architectural configuration and signal conditions are expected to change.

## Figures and Tables

**Figure 1 sensors-25-04092-f001:**
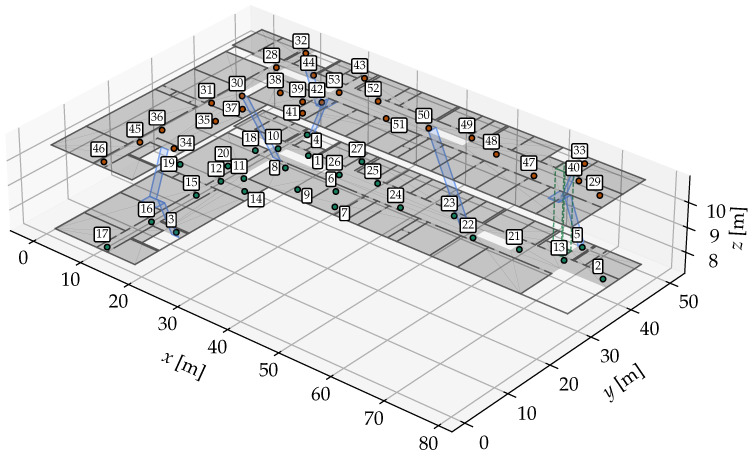
Example environment consisting of two floors of a university building. The green dots mark FPs on the lower floor (r=2), the orange dots on the upper floor (r=3). The FPs are numbered according to the numbers in the white boxes next to the dots. The floors are connected with stairs (pale blue) and an elevator (pale green).

**Figure 2 sensors-25-04092-f002:**
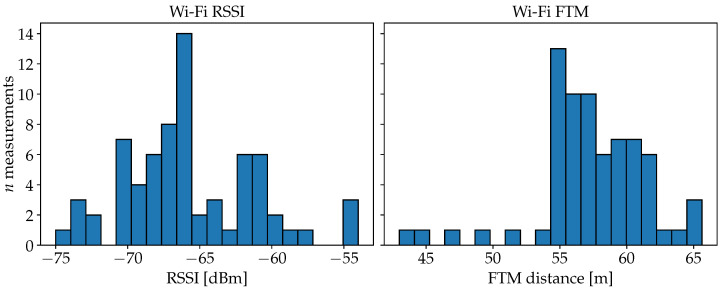
Distribution of Wi-Fi RSSI and FTM values for measurements taken at the same location from an AP placed 10.8 m away in the line-of-sight. Both distributions are also spread out significantly and the FTM distances are also far off from the true distance in this example.

**Figure 3 sensors-25-04092-f003:**
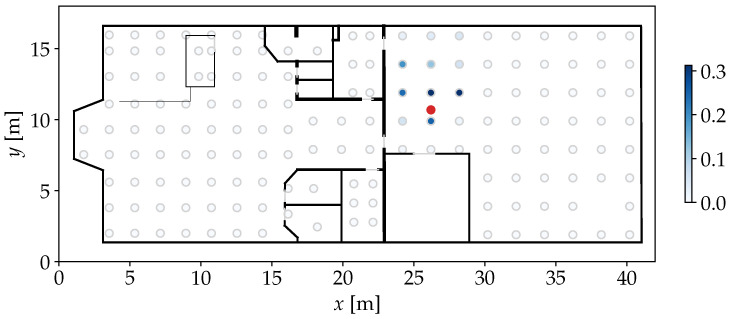
Floor plan of an indoor environment with FP positions marked by light gray outlines. The darkness of the fill color indicates the similarity of each FP to a given test Wi-Fi RSSI observation recorded at the red marker. As intended, similarities are visibly concentrated around the true position.

**Figure 4 sensors-25-04092-f004:**
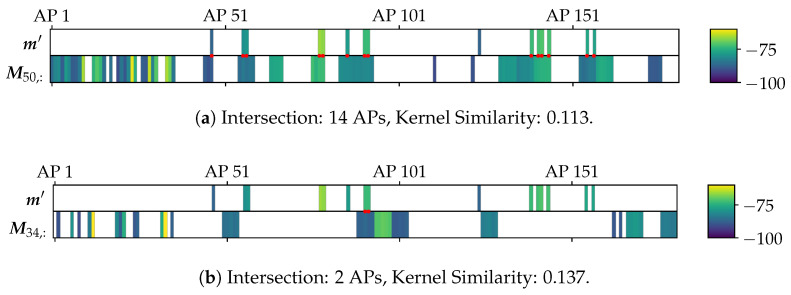
Similarity computation with RBF kernel (l=10) for an observation $m^{\prime }$ with two different FPs, using only elements present in both vectors. Values are the mean aggregations of Wi-Fi RSSI [dBm] per AP from a real-world scenario, showcased in Figure 1. The intersecting features are marked in red. The FP in Figure 4a has a higher intersection and appears visibly similar to $m^{\prime }$. But here, the kernel similarity over the intersecting features is lower than with the FP in Figure 4b with just minimal intersection.

**Figure 5 sensors-25-04092-f005:**
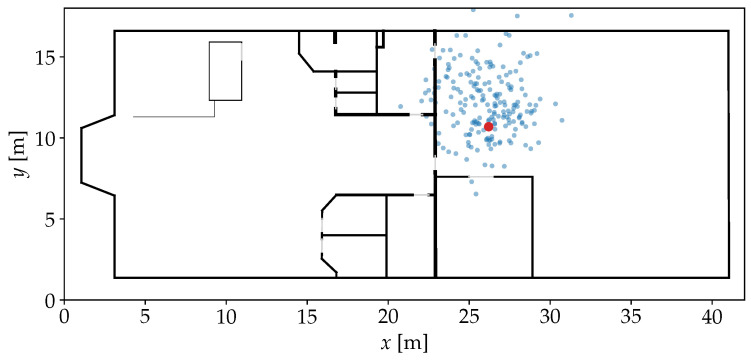
Position samples $\hat{P}$ generated based on the FPs and similarities from Figure 3 with η=500 and $\sigma^{2}=1.0$. The red marker stands for the position where the observation was recorded before. The noise causes the samples to spread out over the area between the FP positions that have a high similarity. The spread also causes a few samples to lay in another room or outside of the building.

**Figure 6 sensors-25-04092-f006:**
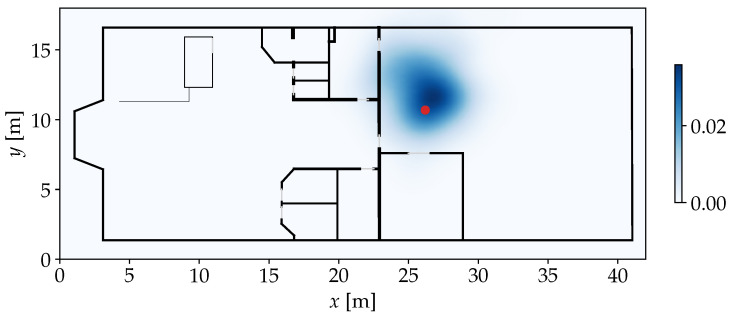
KDE applied on $\hat{P}$ from Figure 5 with h=1. The red marker highlights the true position. The resulting density function (blue) provides a smooth continuous representation of the original similarities over the entire area of the environment.

**Figure 7 sensors-25-04092-f007:**
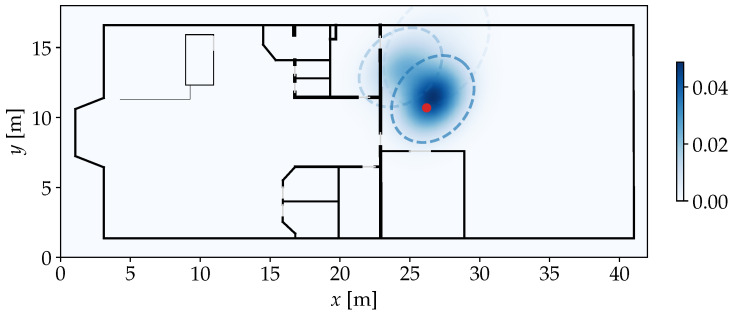
DPGMM applied on $\hat{P}$ from Figure 5. The red marker highlights the true position. The resulting density function is visualized by the blue continuous gradient. Each component is marked by a dashed ellipse with its opacity indicating the weight. Most of the weight is concentrated on just two partially overlapping components.

**Figure 8 sensors-25-04092-f008:**
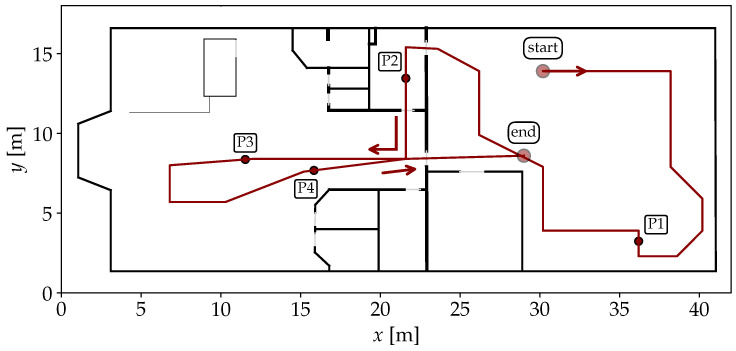
Scenario #1: Floor plan of the combined workshop and office building (GT in red). Start- and endpoint are highlighted and the walking direction is clarified by arrows. P1–P4 mark interesting points on the path during the recording and are referred to in the discussion of the results.

**Figure 9 sensors-25-04092-f009:**
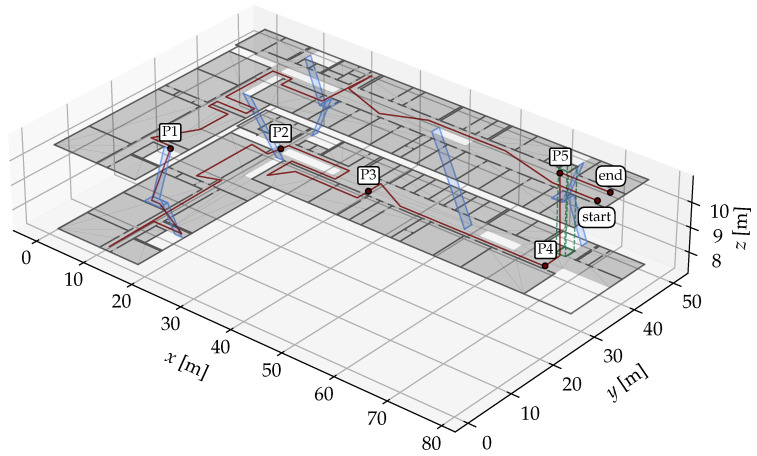
Scenario #2: Floor plan of the university building (GT in red). P1–P5 mark interesting points on the path during the recording and are referred to in the discussion of the results.

**Figure 10 sensors-25-04092-f010:**
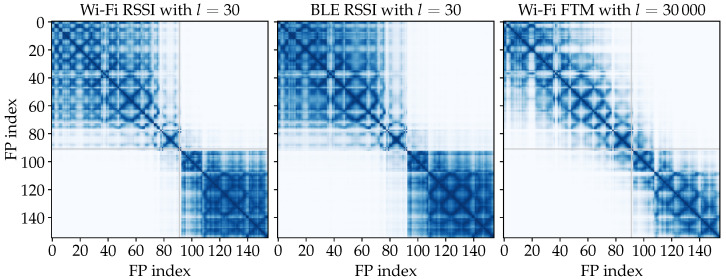
Visualization of pairwise RBF kernel similarities for M with mean-aggregated observations from the first evaluation scenario. *l* is chosen large to provide a comprehensive impression. Wi-Fi and BLE RSSI have almost identical patterns, whereby BLE appears smoother. For FTM, there are still visible patches, but the 2 clearly distinct clusters from RSSI are not as clear here.

**Figure 11 sensors-25-04092-f011:**
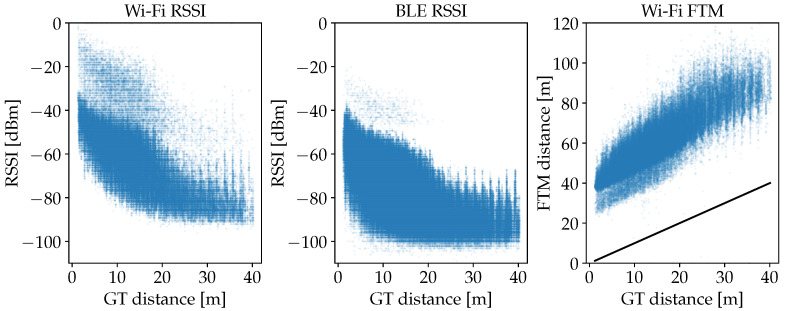
Scatter plots for the correlation of radio measurements (blue) and GT distances from the entire FP data of the first evaluation scenario. For the Wi-Fi and BLE RSSI values, the overall correlation is similar, whereby there are much more datapoints available for BLE. Especially for Wi-Fi, a less dense subset is visible that is seemingly shifted from the rest of the distribution between 0 m and 10 m. For FTM, which provides actual distances, the measurements are more or less in parallel to the ideal GT distances that are given by the straight black line, but with a significant offset of about 50 m.

**Figure 12 sensors-25-04092-f012:**
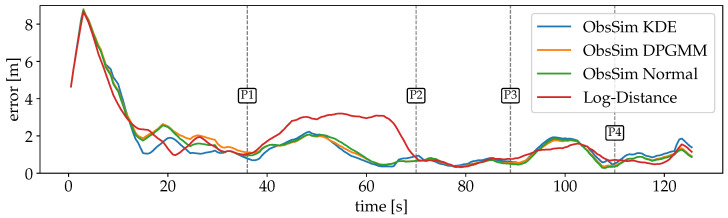
Errors over time for scenario #1 with Wi-Fi for each model. Visualizations are smoothed with convolution to increase clarity. The error courses are mostly similar for every ObsSim configuration. After an initial peak, the errors are concentrated around 2 m and below after the first 17 s. The Log-Distance curve is slightly off, especially between 40 s and 70 s.

**Figure 13 sensors-25-04092-f013:**
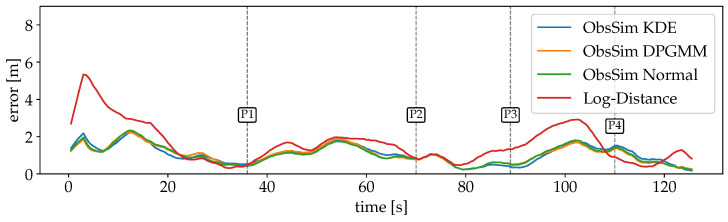
Smoothed errors over time for scenario #1 with BLE for each model. The course of errors is very consistent across all ObsSim configurations. For the Log-Distance model, there are some more intense deviations.

**Figure 14 sensors-25-04092-f014:**
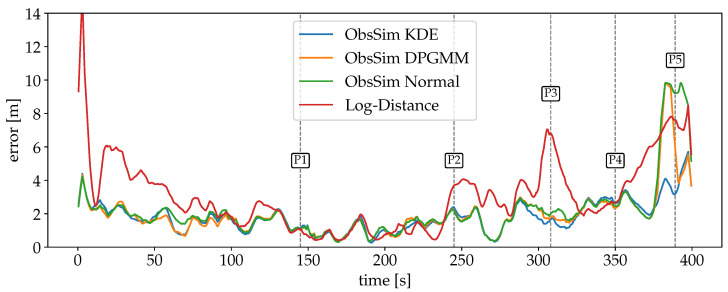
Smoothed errors over time for scenario #2 with Wi-Fi for each model. With no noticeable impact of the stairs at P1, the course of all errors is mostly consistent until P2 despite deviations for Log-Distance. The larger deviations at P2 and P3 for the Log-Distance model appear all on the longer segment of the corridor. The elevator ride, which is marked by P4 and P5, impacts the accuracy of all configurations except for ObsSim with KDE.

**Figure 15 sensors-25-04092-f015:**
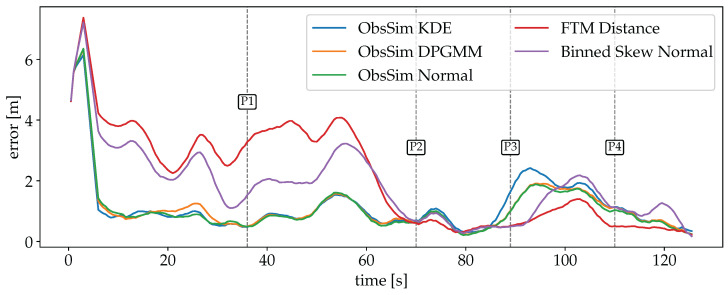
Smoothed errors over time for scenario #1 with FTM for all used models. After an initial peak for all models, there is a clear separation between the plain FTM Distance model, which is the worst, the Binned Skew Normal model, which is notably better, and the ObsSim model that is by far the most accurate. In the second half, all model accuracies are more close to each other and the FTM Distance model is even the best, temporarily.

**Table 1 sensors-25-04092-t001:** Adjusted model hyperparameters. For the ObsSim model, *l* is the kernel length-scale, $\sigma^{2}$ is the sampling noise for generating $\hat{P}$ in case of KDE or DPGMM and normal distribution, respectively. *h* is the bandwidth in case of KDE. For the Log-Distance model, $\sigma^{2}$ is the noise added to decide whether a measured distance fits the actual distance from a well-known position to an AP.

	ObsSim 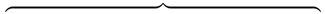	Log-Dist 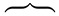
	*l*	$\sigma^{2}$	*h*	$\sigma^{2}$	$\sigma^{2}$
**Scenario**	**all**	**KDE**	**KDE**	**DP./Norm.**	**-**
#1 (Wi-Fi)	3.0	0.5	5.0	5.0	3.0
#1 (BLE)	3.0	0.5	12.0	8.0	5.0
#2 (Wi-Fi)	12.0	0.5	7.0	5.0	7.0

**Table 2 sensors-25-04092-t002:** Experiment error summary for scenario #1 with Wi-Fi for each model. Each value is the average absolute positioning error [m] for each time step.

Result	Mean	Median	75 % Qnt.	90 % Qnt.	Std. Dev.
ObsSim KDE	1.65	1.03	1.86	2.89	1.86
ObsSim DPGMM	1.70	1.23	1.86	2.89	1.83
ObsSim Normal	1.66	1.18	1.86	2.77	1.83
Log-Distance	1.97	1.38	2.71	3.28	1.79

**Table 3 sensors-25-04092-t003:** Experiment error summary for scenario #1 with BLE for each model. Each value is the average absolute positioning error [m] for each time step.

Result	Mean	Median	75 % Qnt.	90 % Qnt.	Std. Dev.
ObsSim KDE	1.07	1.05	1.43	1.82	0.58
ObsSim DPGMM	1.07	1.02	1.38	1.78	0.56
ObsSim Normal	1.06	0.98	1.39	1.78	0.58
Log-Distance	1.64	1.42	2.03	3.10	1.16

**Table 4 sensors-25-04092-t004:** Experiment error summary for scenario #2 with Wi-Fi for each model. Each value is the average absolute positioning error [m] for each time step.

Result	Mean	Median	75 % Qnt.	90 % Qnt.	Std. Dev.
ObsSim KDE	1.82	1.72	2.32	2.97	1.02
ObsSim DPGMM	1.95	1.74	2.28	2.94	1.48
ObsSim Normal	2.16	1.81	2.36	3.00	1.85
Log-Distance	3.13	2.62	3.99	6.25	2.44

**Table 5 sensors-25-04092-t005:** Experiment error summary for scenario #1 with FTM for each model. Each value is the average absolute positioning error [m] for each time step.

Result	Mean	Median	75 % Qnt.	90 % Qnt.	Std. Dev.
ObsSim KDE	1.20	0.88	1.31	2.03	1.38
ObsSim DPGMM	1.16	0.88	1.34	1.80	1.35
ObsSim Normal	1.12	0.86	1.22	1.75	1.36
FTM Distance	2.16	1.72	3.63	4.00	1.82
Binned Skew Normal	1.88	1.73	2.39	3.26	1.52

## Data Availability

Data available on request due to restrictions.

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
