# Peer review of "Unified Probabilistic and Similarity-Based Position Estimation from Radio Observations"

_sensors, 2025, doi:10.3390/s25134092_

Round 1

Reviewer 1 Report

Comments and Suggestions for Authors

Please, in order to clarify the paper you should resume it, specifically the part of the state of the art.

In order to clarify the paper, I recommend you to follow the typical sections of a research paper, Introduction, Methods and Materials, Results, Discussion, Conclusions and future work.

Reorder the paper and introduce the section of resumed section of the state of the art in the Introduction Section. Now, the paper is not clear ant it is too long.

MINOR QUESTIONS

In the abstract correct “assinment”.

About acronyms, if the terms abbreviated con an acronym is only used once in the abstract, the acronym is not necessary.

In acronyms are needed, acronyms have to appear in the abstract and the main text

What is the meaning of “FTM”.

More details:

  • The paper provides a technique of location in indoor environments.
  • The scope of the paper is not new.  There is many literature about indoor location.
  • The paper is very confusing.
    It is too large. Authors should resume the state of the art. The paper should be rewrite- The English is not quite good.
    Authors should rewrite thinking the mean idea that they what to transmit. For example, the abstract is chaotic. It has no sense. In the abstract, authors should remark the resume of the work, with the mean findings, and that is all. I think everything is a bit confused.
    I have the sense that authors have started to write, they continue… writing…, and that’s all, without thinking of a propose. Writing for the sake of writing is meaningless
    I have the sense that the abstract has been written before writing the complete paper
  • Think what the purpose of the work, and the purpose of each section, and then start to write. And then review the English, very important.
    Please respect the typical sections of a research paper.

Please remember: Good things, if brief, are twice as good.

I would change the title of the paper.

  • The Conclusions are a bit better. This section reflects the validity of the proposed model, and why it is good, and the comparison with other techniques is shown. And then the future work. The Conclusion section is more coherent.
  • The references are appropriate.
  • Tables and Figure are correct.
    Figures are very clear, all the physic magnitudes are indicated properly, and the units are also correct. The figures are quite good, the graphs in three dimensions are very clear. The quality of the figures is quite good. The colors are well chosen.
Comments on the Quality of English Language

The work should be review by a native/teacher english

Author Response

Comments 1: Please, in order to clarify the paper you should resume it, specifically the part of the state of the art.
Response 1: We always take comments on the preparation of the state of the art very seriously. We therefore generally welcome critical comments. We have now tried to make the three areas of the state of the art more clear by pointing out the purpose of each part of the section in relation to our own work. l.71-77
___ Comments 2: In order to clarify the paper, I recommend you to follow the typical sections of a research paper, Introduction, Methods and Materials, Results, Discussion, Conclusions and future work.
Reorder the paper and introduce the section of resumed section of the state of the art in the Introduction Section. Now, the paper is not clear ant it is too long.
Response 2: Thank you for the suggestions. We considered them by integrating state of the art into the introduction and reorganizing the remaining sections. We are now close to your recommended sections but we still find that it is necessary in our case to distinguish between existing work on which our work is based and the actual proposed methodology.
___ Comments 3: In the abstract correct “assinment”.
About acronyms, if the terms abbreviated con an acronym is only used once in the abstract, the acronym is not necessary.
In acronyms are needed, acronyms have to appear in the abstract and the main text
What is the meaning of “FTM”
Response 3: We agree to all of these suggestions and considered them in the revised manuscript.
___ Comments 4: The paper provides a technique of location in indoor environments. The scope of the paper is not new. There is many literature about indoor location. The paper is very confusing. It is too large. Authors should resume the state of the art. The paper should be rewrite- The English is not quite good.
Response 4: Thank you, we were indeed aware that we were caught between established techniques and a paper with a still appropriate extent. However, having clarified the structure of the paper, we believe that experienced readers will find it easier to skip over familiar sections. We would be reluctant to remove the explanations, which have made the paper quite extensive, as important information might be lost for less experienced readers. Regarding English writing, we improved the entire paper.
___ Comments 5: Authors should rewrite thinking the mean idea that they what to transmit. For example, the abstract is chaotic. It has no sense. In the abstract, authors should remark the resume of the work, with the mean findings, and that is all. I think everything is a bit confused. I have the sense that authors have started to write, they continue… writing…, and that’s all, without thinking of a propose. Writing for the sake of writing is meaningless I have the sense that the abstract has been written before writing the complete paper Think what the purpose of the work, and the purpose of each section, and then start to write. And then review the English, very important. Please respect the typical sections of a research paper. Please remember: Good things, if brief, are twice as good.
Response 5: Thank you for pointing out the issues with the abstract. We have revised it to highlight the essential aspects of our work. As mentioned in earlier responses, we acknowledge your feedback and improved English writing and the structure of the paper.
___ Comments 6: I would change the title of the paper.
Response 6: We appreciate the feedback and welcome further input. After extensive discussion, we believe that the alternative title "Unified Probabilistic and Similarity-based Position Estimation from Radio Observations" is also appropriate to convey the core ideas of the paper. If the editor would like to contribute an opinion on the title, we would be open to considering it.
___ Comments 7: The Conclusions are a bit better. This section reflects the validity of the proposed model, and why it is good, and the comparison with other techniques is shown. And then the future work. The Conclusion section is more coherent. The references are appropriate. Tables and Figure are correct. Figures are very clear, all the physic magnitudes are indicated properly, and the units are also correct. The figures are quite good, the graphs in three dimensions are very clear. The quality of the figures is quite good. The colors are well chosen.
Response 7: Thank you for highlighting the positive aspects. This is also valuable feedback.

Reviewer 2 Report

Comments and Suggestions for Authors

Dear Authors

I have reviewed your paper with interest.
It is very well written and is almost ready for acceptance.
However, there are two points that concern me. Please respond to these.

1) There is no space between Equation 7 and the text below it. Please leave an appropriate space.

2) Figure 7 is not mentioned in the text. Please mention it in the text.

Sincerely yours,

Author Response

Comments 1: I have reviewed your paper with interest. It is very well written and is almost ready for acceptance.
Response 1: Thank you very much for your kind words and positive feedback on our paper. We appreciate your review and will make the suggested revisions promptly to ensure the work meets high standards.
___ Comments 2: There is no space between Equation 7 and the text below it. Please leave an appropriate space.
Response 2: Thank you for pointing this out. We checked the passage and confirm that it is indeed technically correct to leave no space here, since the sentence, in which the equation is embedded, is continued below. However, we identified an unnecessary line break above the equation, which caused the spacing to appear "unbalanced". We corrected that.
___ Comments 3: Figure 7 is not mentioned in the text. Please mention it in the text.
Response 3: Indeed, that was a mistake on our side. We are glad you noticed it and added a reference where it should belong. ll.376-377

Reviewer 3 Report

Comments and Suggestions for Authors

The contribution of this article is to propose an indoor positioning method based on Wi-Fi RSSI and FTM data, and optimize it for the positioning problem of mobile pedestrians. First, the article introduces an observation model based on feature aggregation and uses kernel functions to calculate the similarity of observation vectors, which helps to reduce the data sparsity problem in the localization process. Secondly, the article proposes a particle filter (PF)-based method to process dynamically updated observation data, and on this basis performs position density estimation, improving positioning accuracy by weighting and filtering samples. In particular, the authors used kernel density estimation (KDE) to smooth the sample distribution, providing a more stable and continuous probability distribution for the localization results. Overall, this method can effectively deal with the multipath effects of radio waves and environmental changes in actual scenarios, improve positioning accuracy and stability, and has important reference value for the development of indoor positioning technology.

There are a few issues that require the author's assistance, which will allow us to learn.

1. This problem aims to explore the main differences between similarity-based methods and distance-based models, with a focus on the performance of the method in terms of accuracy, and how its flexibility in handling different types of signal measurements (such as RSS and FTM) can improve the accuracy of pedestrian tracking. How does the proposed similarity-based approach compare to traditional distance-based models in terms of accuracy and flexibility?

2. The advantages and challenges of using kernel-based similarity measures, especially how to maintain the accuracy and stability of position estimates in environments with highly dynamic or unstable signal propagation conditions. What are the advantages and limitations of using kernel-based comparison methods to match live data to reference recordings at known locations? How does this method perform in environments with widely varying signal conditions?

3. How this similarity-based model copes with the irregularities in signal propagation caused by obstacles such as walls and furniture, and its advantages in handling uncertainty and improving positioning accuracy compared to traditional positioning methods that rely on physical assumptions. How can similarity-based methods effectively handle noise and irregularities in signal measurements, and what advantages do they have over traditional positioning methods based on physical assumptions?

4. Challenges that probabilistic models of observational similarity may encounter in practical use, such as how the model can maintain stability and accuracy and provide valid estimates when the signal source changes or objects move in indoor environments. This method uses a probabilistic model based on observation similarity to estimate the location. What challenges does such a model face in practical applications, especially in dynamic or changing environments?

5. How the similarity localization model copes better with signal changes and interference than the traditional distance model, and the challenges it may face in dynamic or complex indoor environments. Compared with traditional distance-based localization methods, how can similarity-based localization models handle irregular signal propagation and environmental noise more effectively? What are the potential challenges in applying this approach in practice?

6. Compare the fusion of similarity-based and distance-based methods in localization systems and explore the relative advantages and limitations of these methods in implementation, especially in dealing with high uncertainty and dynamically changing environments. When performing positioning fusion, what are the similarities and differences between similarity-based methods and distance-based methods? What are the advantages and limitations of the application of similarity models in the fusion process compared with traditional distance models?

7. The advantages of the ObsSim model in flexibility and accuracy over traditional RSSI or FTM models, and explore whether it faces challenges or performance bottlenecks in specific environments or scenarios. In what ways can the similarity-based ObsSim model provide more flexible and accurate indoor positioning compared to the traditional RSSI or FTM models? Will there be performance bottlenecks in certain specific environments?

8. When integrating the ObsSim model into an existing indoor positioning system, how do you ensure that it works in conjunction with other sensors (such as IMU)? How does the model handle possible missing data or incomplete measurements?

9. The article mentioned that when performing density estimation, Gaussian kernel function was used for weighting. Will this choice affect the positioning accuracy? If so, how to choose a suitable kernel function to improve the localization results?

Comments on the Quality of English Language

Please ask the author to check the grammar again.

Round 2

Reviewer 1 Report

Comments and Suggestions for Authors

The paper could has been summarized, but I am going to accept it.

Author Response

Thank you.

Reviewer 3 Report

Comments and Suggestions for Authors

I have reviewed the paper, and the author has adequately responded to the issues that require revision. 

Author Response

Thank you.